# Determination of the Nutritional Value of Diet Containing *Bacillus subtilis* Hydrolyzed Feather Meal in Adult Dogs

**DOI:** 10.3390/ani11123553

**Published:** 2021-12-14

**Authors:** Geruza Silveira Machado, Ana Paula Folmer Correa, Paula Gabriela da Silva Pires, Letícia Marconatto, Adriano Brandelli, Alexandre de Mello Kessler, Luciano Trevizan

**Affiliations:** 1Programa de Pós-Graduação em Zootecnia, Universidade Federal do Rio Grande do Sul, Porto Alegre 91540-000, RS, Brazil; gsm_sg@hotmail.com (G.S.M.); paulagabrielapires@yahoo.com.br (P.G.d.S.P.); akessler@ufrgs.br (A.d.M.K.); 2Programa de Pós-Graduação em Recursos Naturais, Universidade Federal de Roraima, Boa Vista 69304-000, RR, Brazil; folmercorrea@gmail.com; 3Instituto do Petróleo e dos Recursos Naturais, Pontifícia Universidade Católica do Rio Grande do Sul, Porto Alegre 90160-091, RS, Brazil; leticiamarconatto@gmail.com; 4Programa de Pós-Graduação em Ciência e Tecnologia de Alimentos, Universidade Federal do Rio Grande do Sul, Porto Alegre 91509-900, RS, Brazil; abrand@ufrgs.br

**Keywords:** feather processing, total tract digestibility, hydrolyzed protein, poultry byproduct

## Abstract

**Simple Summary:**

The production of meat for human consumption produces extra ingredients used in animal nutrition. Feathers, for example, account for about 7% of the chicken’s body weight. When discarded, it presents a potential risk of environmental contamination. Feathers are minimally digested in mammals and are a very rich source of protein. Improved digestibility can be done by thermal processing or by microorganisms. *Bacillus subtilis* was shown to have great feather-degrading activity In vitro and we produced an amount of microbial hydrolysate to test in dogs. We did some evaluations on the ingredient to measure the effects of the microorganism on feathers. In dogs, a test of total tract digestibility, microbial resistance to the gastrointestinal tract, and fecal characteristics were performed. *Bacillus subtilis* was less efficient to digest feathers when a higher concentration of feathers was added to the culture. The amino acid profile in feathers has probably changed due to fermentation. Dogs ate the diets quickly, with no refusals. Nutrient and energy total tract digestibility were lower when compared to thermally processed feathers, but *Bacillus subtilis* was found viable in the feces of dogs that ingested fermented feathers, signaling that *Bacillus subtilis* is resistant to digestion and may bring some probiotic effect.

**Abstract:**

Feathers are naturally made up of non-digestible proteins. Under thermal processing, total tract digestibility can be partially improved. Furthermore, *Bacillus subtilis* (Bs) has shown a hydrolytic effect In vitro. Then, a Bs FTC01 was selected to hydrolyze enough feathers to produce a meal, and then test the quality and inclusion in the dog’s diet to measure the apparent total tract digestibility coefficient (ATTDC) in vivo and the microorganism’s ability to survive in the gastrointestinal tract. A basal diet was added with 9.09% hydrolyzed Bs feather meal (HFMBs) or 9.09% thermally hydrolyzed feather meal (HFMT). Nine adult dogs were randomized into two 10-day blocks and fed different diets. Microbial counts were performed on feather meal, diets and feces. The Bs was less effective in digesting the feathers, which reduced the ATTDC of dry matter, crude protein, energy and increased the production of fecal DM, but the fecal score was maintained (*p* > 0.05). The digestible energy of HFMT and HFMBs was 18,590 J/kg and 9196 J/kg, respectively. *Bacillus subtilis* showed limitation to digest feather in large scale, but the resistance of Bs to digestion was observed since it grown on feces culture.

## 1. Introduction

In poultry production, feathers are considered a by-product of slaughterhouses and correspond to about 7% of the bird’s body weight [1]. According to FAO [2], a total of 114,267 thousand tons of broiler chicken meat was produced in the world, and estimated feather production was close to 8000 thousand tons in the same year. In the industry, feathers are transformed into hydrolyzed feather meal, through digestion by thermal processing, with high pressure and temperature, generating an ingredient with high protein concentration. Feather meal normally contains more than 85% of crude protein in dry matter, mostly from keratin, the main constituent of the feather. However, the method is not satisfactory, since the in vivo total tract digestibility of feather protein is relatively low, around 60% [3,4]. Specifically, low total tract digestibility values of about 67% have been reported in dogs [5].

The use of proteolytic microorganisms for feather hydrolysis can be an alternative to the traditional method, since its effect on nutrient availability is real. According to Cedrola [6], the biodegradation of keratin by microorganism represents an alternative method to improve the nutritional value of feathers, improving the quality of the protein that has low total tract digestibility, even if there is loss of nutrients during the fermentation. The increased availability of amino acid (AA) improves the ingredient value and permit incorporation in diets for different species in greater proportion, especially as a source of sulfur AA. Another important factor to be considered is the low molecular weight acquired by the soluble portions of the feather meal processed by heat and pressure [5]. The association between microorganisms and thermal processing can improve the availability of AA, once more. Currently, protein ingredients with such characteristics have potential use in hypoallergenic diets for dogs and cats, a category of pet food in high demand [7,8].

Another interesting point is the accumulation of microorganisms in the feather meal after processing, when beneficial microbes are used for hydrolysis. Some strains of *Bacillus* spp. are recognized as feather-degrading bacteria and to exert beneficial effects on the host health after oral administration [9,10]. The sporulated forms has the ability to resist the low stomach pH and can reach the intestine in large quantities, where they can produce competitive exclusion with non-desired bacteria which is known as a probiotic effect [11]. The aim of the present study was to compare the thermally hydrolyzed feather meal with *Bacillus subtilis* hydrolyzed feather meal to determine the quality of the ingredient and total tract digestibility of nutrients and energy in diets for adult dogs. In addition, the research seeks to highlight the potential of *B. subtilis* resisting to digestion, then it could be studied as a probiotic agent in dogs.

## 2. Materials and Methods

All animal care and handling procedures were approved by the Institutional Animal Care and Use Committee at the Universidade Federal do Rio Grande do Sul, protocol number 20643.

### 2.1. Animals

Nine healthy adult Beagle, coming from Animal Science Department, Universidade Federal do Rio Grande do Sul, Porto Alegre, Brazil, were used in this study. They were all intact, between 2 and 3 years old, weighing 11.3 ± 1.60 kg, and free of endo and ectoparasites. All dogs were regularly immunized and submitted to clinical and laboratory tests to measure complete blood count (CBC) and to perform biochemical and coproparasitological analyses before starting the study. The dogs were housed in individual stainles steel metabolic cages (1.0 m × 1.0 m × 1.5 m) equipped with a feces and urine collector, feeders, and drinkers in a controlled room at 25 °C, with a light dark cycle of 14 h:10 h.

### 2.2. Diets

Two feather meals were produced to compare the thermal and microbial fermentation on feather degradation and nutrient availability (Table 1).

The experimental diets were produced from an extruded basal diet, formulated to meet the energy and nutrient requirement of adult dog, as recommended by the FEDIAF [12]. Basal diet was replaced with 9.09% *B. subtilis* hydrolyzed feather meal or 9.09% thermally hydrolyzed feather meal to composed the treatments (Table 2).

**Table 1 animals-11-03553-t001:** Chemical composition and amino acid profile of hydrolyzed feather meal by *Bacillus subtilis* (HFMBs) and thermally hydrolyzed feather meal (HFMT) expressed in dry matter basis.

Itens, % DM Basis	HFMBs (%)	HFMT (%)
Dry matter	93.6	93.7
Crude protein	88.3	84.1
Fat	6.80	8.30
Mineral matter	3.20	2.90
Gross energy, J/kg	23,033	24,552
Amino acids profile, %		
Aspartic acid ^1^	6.95	6.23
Glutamic acid ^1^	10.8	9.06
Serine ^1^	10.2	10.9
Glycine ^1^	7.62	6.47
Histidine ^1^	0.85	0.41
Arginine ^1^	6.25	5.67
Threonine ^1^	3.93	3.99
Alanine ^1^	4.01	3.57
Proline ^1^	8.44	9.87
Tyrosine ^1^	2.43	2.11
Valine ^1^	6.12	6.58
Methionine ^1^	1,53	2.41
Cystine ^1^	3.38	8.11
Isoleucine ^1^	4.10	4.29
Leucine ^1^	7.31	6.98
Phenylalanine ^1^	4.25	4.39
Lysine ^1^	2.43	1.14
Tryptophane ^2^	0.40	0.19
Total aas	91.0	92.4

^1^ HPLC, according to White et al. [13] and Hagen et al. [14]. ^2^ Hydrolysis and determination, according Lucas & Sotelo [15].

#### 2.2.1. Thermally Hydrolyzed Feather Meal (HFMT)

Thermal hydrolysis was used to produce HFMT in an industrial plant. Feathers from the slaughterhouse were visually inspected upon discharge and then received the first addition of antioxidant to protect the raw material during thermal hydrolysis. The process consists of a temperature ramp from 110 °C to 170 °C during 20 min, reaching 303.9 kPa pressure (Kabsa Exportadora S.A., Porto Alegre, RS, Brazil). The product was discharged into the continuous flow dryer where it remained for 30 min. The product was maintained at 110 °C with a gradual drop to 50 °C. After drying, the feathers were sieved, ground, and received an additional antioxidant addition before being bagged. The production was monitored and about 50 kg of the meal originated from the same batch was separated to carry out the evaluation.

#### 2.2.2. Bacillus Subtilis Hydrolyzed Feather Meal (HFMBs)

HFMBs was produced in the laboratory. The *B. subtilis* FTC01(9) strain used in the study was multiplied to form the pre-inoculum in brain heart infusion broth (BHI; Himedia, Mumbai, India) incubated for 24 h at 37 °C, with agitation of 120 rpm. The pre-inoculum was added to a broth (0.5 g/L NaCl, 0.3 g/L K_2_HPO_4_, 0.4 g/L KH_2_PO_4_) containing 10% (*w*/*v*) feathers, and incubated in a bioreactor with agitation at 70 rpm at 37 °C, for 120 h. Then, the digested product was placed in plastic forms composing a 2 cm deep film, and submitted to drying in a forced air oven at 55 °C for 72 h. The dried material was ground through a 1-mm screen in a Wiley mill (DeLeo Equipamentos Laboratoriais, Porto Alegre, Brazil). Five baths in a bioreactor were made to achieve the mass of hydrolyzed feathers sufficient to make the meal necessary to carry out in vivo evaluation. At the end, all baths were homogenized to form a single meal.

##### Evaluations on the Feather Processing

Degradation Factor

Feathers processed by microorganisms were subjected to degradation factor assessment. The material was filtered through a 2.5 mm screen filter and the non-degraded feather residues were washed with distilled water, dried in an oven, and weighed. This factor aims to estimate the degree of degradation from the original content according to the following formula:Degradation factor = [(initial feather mass − final feather mass)/(initial feather mass)](1)

Ultrastructural Evaluation of Feathers

To verify the action of the processing on the structure of the feathers, samples of the feathers in natura and the processed meal were analyzed by scanning electron microscopy. The samples were mounted on a stub, coated with gold-palladium of 35 nm for 3 min (Sputter Coater-SCD 050 Balzers, Germany), and analyzed through a scanning electron microscope (JEOL 6060, Tokyo, Japan) at a standard magnification of 500×. The integrity of the structure of the thermally processed feathers and those processed by microorganisms were observed.

Aminograms and Protein Efficiency Ratio (PER)

The aminograms were performed using high performance liquid chromatography (HPLC) following the methodology described by White et al. [13] and Hagen et al. [14]. Tryptophan was analyzed according to the methodology described by Lucas & Sotelo [15]. The protein efficiency ratio value was calculated and compared with in natura feather, from the composition of the AAs found in the aminograms, based on an equations developed by Alsmeyer, Cunningham & Happich [16], as follow:PER = − 1.816 + 0.435 (methionine) + 0.780 (leucine) + 0.211 (histidine) − 0.944 (tyrosine)(2)

Microbial Count (Colony Forming Unit—CFU)

The counting of microorganisms was performed by the surface counting technique, after addition of 25 g feather meal in 225 mL of saline solution (NaCl, 0.85% *w*/*v*), followed by serial dilutions, adding 1 mL of each dilution into tubes containing 9 mL of saline solution. An aliquot of each dilution (0.1 mL) was transferred to Petri dishes containing Nutrient Agar (20 g/L agar-agar and 13 g/L Nutrient Broth) as culture medium. Subsequently, the aliquot was spread over the medium with the aid of a Drigalsky loop until the aliquot was completely adsorbed onto the medium. The Petri dishes were incubated at 37 °C for 24 h. Colonies were identified by morphology and counted.

### 2.3. Experimental Diets and Feed Management

The experimental diets were prepared using a basal diet (BD) that was added with 9.09% (*w*/*w*) HFMBs or 9.09% (*w*/*w*) HFMT according to the methodology described by Adeola [17]. The feather meal was added over the diet directly in the feeders, previous to offer to the dogs. Three treatments were acquired: BD, BD + HFMBs (9.09%) and BD + HFMT (9.09%). Meals were calculated based on the BD according to the needs of each dog to reach the maintenance requirements established by NRC [18] (Table 2). The ME (J/kg) of BD was estimated by the equations proposed by the NRC [18] and the amount of food supplied to the dogs was calculated according to the maintenance energy requirement of adult dogs [ME = 502 J × body weight (kg)^0.75^/day]. The amount of food was divided into two equal portions and provided at 09:00 a.m. and 05:00 p.m. Water consumption was measured individually, and evaporative losses were discounted by an intact drinker located into the room.

### 2.4. Apparent Total Tract Digestibility (ATTD) Assay

#### 2.4.1. Experimental Design

The experiment was carried out in a randomized block design consisting of two blocks of 10 days, with three treatments and three dogs per treatment in each block, totaling six replications per treatment. Each period included a five-day adaptation phase (day 1 to day 5) followed by a phase of total collection of feces and urine (day 6 to day 10). At the beginning and at the end of each collection period, a gelatin capsule containing 1000 mg of iron (III) oxide (Fe_2_O_3_) was given orally to separate the collection periods. Feces were collected and stored under −20 °C until the analysis. Body weights (BW) were obtained in the day 1 and 10.

#### 2.4.2. Fecal Score Assessment

During data collection, the fecal score was measured in each defecation according to the following scale: 1 = dry and hard; 2 = well formed, leaves no mark when picked up; 3 = wet start to lose shape, leaving a definite mark when removed; 4 = most, if not all, shape is lost, weak, viscous consistency and 5 = watery diarrhea [19].

#### 2.4.3. Urinary Analysis

During collection, the total volume of urine was measured with the aid of a graduated cylinder and an aliquot was taken daily to measure urine pH and urine density. Digital pH meter (Kasvi modelo K39-2014B, São José dos Pinhais, PR, Brazil) previously calibrated with pH 7 and 10 buffer solutions was used to measures pH and density was measured with a portable refractometer (BEL RPI 89/336, Guangzhou, China).

### 2.5. Chemical Analysis

Stool from each dog was thawed, homogenized, and dried in a forced air oven at 55 °C for 72 h, according to the recommendations of the Association of Official Analytical Chemists [20]. Feces, feather, and diets were ground through a 1 mm screen in a Wiley hammer mill (DeLeo Equipamentos Laboratoriais, Porto Alegre, Brazil), and analyzed for dry matter (DM—AOAC 934.01), mineral matter (MM—942.05), organic matter (OM—920.36) ether extract (EE—AOAC 954.02; model 170/3, Fanem, São Paulo, Brazil), crude protein (CP -AOAC 954.01; model TE 036/2, Tecnal, Piracicaba, Brazil), and 100 mL urine samples were lyophilized for GE analysis following the protocols of the AOAC [20]. The gross energy (GE) of diets and feces were determined using isoperibolic bomb calorimetry (calorimeter, basic Model C2000, Ika^®^-Werke, Staufen, Germany). All analyzes were performed in duplicate and repeated when the variation was greater than 1% for energy and more than 5% for other analyzes (Table 1 and Table 2).

### 2.6. Fecal Microbial Count, pH and Ammonia

Aliquots of feces were collected during the five days to check the fecal microbial count, pH, and fecal ammonia. The microbial counting methodology was the same used in the feather meal. A digital pH meter was used to measure pH (Kasvi model K39-2014B, Paraná, Brazil) previously calibrated with pH 7 and 10 buffer solutions. The ammonia concentration was determined in 3 g of feces, which was incubated for 1 h in a 500 mL Erlenmeyer, containing 250 mL of distilled water. Then, 3 drops of octyl alcohol (1-octanol) and 2 g of magnesium oxide were added to the solution. Then the solution was distilled in a macro-Kjeldahl apparatus and recovered in a flask containing 25 mL of boric acid. Finally, the ammonia was titrated, using standardized sulfuric acid at 0.1 N. The fecal ammonia concentration was calculated as [21]:N-ammonia (g/kg) = N × correction radius × 17 × (blank acid volume)/sample weight (g)(3)

The ammonia concentration in the feces was adjusted for fecal DM. Microbial counts were made in feces and BD by microbial counting technique.

### 2.7. Feces Collection for Microbiota Identification

After five days of adaptation to the diet, 10 g of feces were collected, homogenized aseptically, within 30 min after defecation. The content was divided into three sterile plastic tubes, sealed with parafilm, and kept in a freezer at −80 °C for the fecal microbiota analysis using amplicon sequencing.

#### DNA Extraction, Amplicon Sequencing and Analysis

Before DNA extraction, 3 g of feces from each dog in the treatment were mixed and sampled out to perform DNA extraction. Total DNA was extracted using DNeasy PowerSoil Kit (Qiagen) following the manufacturer’s instructions. To characterize the prokaryotic population present in samples, fragments of the 16S rRNA gene were amplified using universal primers 515F and 806R [22]. Amplification was performed in a 50 μL mixture, consisting of 1.5 mM MgCl_2_, 0.2 μM of each primer, 0.2 mM of each dNTP, 1U Platinum Taq DNA polymerase, 1X PCR reaction buffer, and approximately 10 ng of genomic DNA. PCR conditions used were one initial denaturation step at 95 °C for 3 min, 25 cycles including denaturation for 30 s at 95 °C, annealing for 1 min at 52 °C, and extension for 1 min at 72 °C, with one final extension step for 7 min at 72 °C. After purifying PCR amplicons using Agencount AMPure Beads (Beckman Coulter, Brea, CA, USA), library construction was performed as described in the Ion Plus Fragment Library from an initial amount of 100 ng of DNA. Since all samples were sequenced in a multiplexed PGM run, barcode sequences were used to identify each sample from the total sequencing output. Amplicon sequencing was conducted on an Ion Torrent PGM System (Thermo Fisher, Waltham, MA, USA) using an Ion 316 chip, following the manufacturer’s instructions. Sequences were trimmed to retain only reads longer than 100 bp and high-quality bases (Phred score > 30) using PRINSEQ [23]. Sequences were clustered using UCLUST on QIIME v1.9.1 [24]. OTUs were selected based on 97% sequence similarity and taxonomic data were achieved through the classification algorithm using the GreenGenes 13.8 [25].

### 2.8. Calculation and Statistical Analysis

Digestible energy (DE; J/kg) of hydrolyzed feather meals were calculated according to the methodology described by Kong and Adeola (17).
ATTD_ingredient_, % = [ATTD_test diet_, % − (100% − X%) × ATTD_basal_, %)]/X%(4)
where ATTD_ingredient_ is the ATTD of energy in the test ingredient (%), ATTD_test diet_ is the ATTD of the energy in the test diet (%), ATTD_basal diet_ is the ATTD of the energy in the basal diet, and X is the proportion that the basal diet was replaced by the test ingredient. The data means were analyzed by ANOVA, using Minitab^®^ (2013) and the means compared by the Tukey test (*p* < 0.05). Faecal score values were analyzed using the Kruskal-Wallis non-parametric test (*p* < 0.05).

## 3. Results

The total feather degradation factor was calculated and resulted in 0.45. Evaluation by electron microscopy revealed differences between processed feather meal and in natura feathers (Figure 1).

The original feather structure was altered in the meal processed with microorganisms, and small fragments could be observed with recognizable portions of the whole feather. Through thermal processing, the feather structures have become smaller, and the original feather pattern can no longer be recognized with small globular fragments. Analyzes of HFMT reveal similar concentrations of macronutrients with a greater concentration of mineral matter in the HFMBs (Table 1).

The original AA profile of the feathers was modified by bioprocessing (Table 1). The lysine and tryptophan content in HFMBs were about 100% higher compared to HFMT, but sulfur AAs were reduced. The results of PER were estimated using the AA composition and it resulted in similar values between the ingredients (HFMBs = 2.8; HFMT = 2.7). Culture in plates permitted to identify the growth of *Bacillus* spp. in the HFMBs through the observation of typical *Bacillus* colony. Plates inoculated with feathers and thermally processed feather meal did not show the growth of microorganism.

The tested diets were isoenergetic and isonutritive, except for the protein that varied according to the inclusion of feather meal (Table 2). During the whole experiment, there were no cases of food refusal or episodes of diarrhea, emesis, or any clinical manifestation related to changes in the gastrointestinal tract or another system. During the total tract digestibility test, dogs’ body weight did not change (*p* > 0.05) (Table 3). Dogs consumed all the meals, and no leftovers were observed along the entire experimental assay. The consumption of energy and nutrient were similar, except for the consumption of CP, which was higher for treatments with the inclusion of HFMBs and HFMT (Table 3). Water consumption and urinary characteristics, pH, and density were not altered by the inclusion of HFMBs and HFMT (*p* > 0.05). Fecal characteristics such as fecal score and pH were not affected by processed feather meal (*p* > 0.05). The total production of feces (g/day) was greater in HFMBs treatment (Table 3). The ammonia concentration was also affected by the inclusion of the meals and the greatest concentration was observed on inclusion of HFMT (*p* < 0.05).

HFMBs showed lower ATTD of energy (DE) compared with HFMT (*p* < 0.05). HFMT showed a greater ATTD of energy of 76%. The DE values were 9196 J/kg and 18,589 J/kg for HFMBs and HFMT, respectively (*p* < 0.05) (Figure 2).

Bacterial counts were below the detection limit for BD and HFMT diets but reached more than 10^4^ CFU/g in HFMBs formulation. In feces samples, a greater quantity of typical colonies of *Bacillus* spp. were observed for dogs that received HFMBs, showing counts ranging from 10^5^ to 10^7^ CFU/g (Table 4).

It was possible to observe a remarkable prevalence of phylum Firmicutes in the HFMBs, indicating that *Bacillus* remains in the HFMBs. However, the predominant bacterial phyla observed in the feces of dogs fed both feather meals were similar (Figure 3).

Seven bacterial phyla were detected in animal feces, including: Actinobacteria, Bacteroidetes, Deferribacteres, Firmicutes, Fusobacteria, Proteobacteria, and Tenericutes. The phylum Bacteoidetes was present in both treatments, with a greater abundance of the genus *Bacteroides* (Figure 4). In addition, the presence of two important genera for animal health, *Lactobacillus* (3.7%) and *Escherichia* (3%) were observed in the feces of dogs fed with HFMBs.

## 4. Discussion

The digestion of feathers by microorganisms has been studied for several years. More specifically, many bacterial strains showed an excellent potential to be used in feather-degrading bioprocesses [9,24,26,27,28]. In vitro tests permitted to establish submerged cultivation protocols containing 2% (*w*/*v*) feathers and reaching complete hydrolysis in 48 h [9,29,30]. In our study, 2% (*w*/*v*) feather was considered too low to produce the amount of hydrolyzed feathers needed to conduct the in vivo ATTD study, once the process yield, in terms of DM, would be very low. To improve the yield, we used 10% (*w*/*v*) feathers In vitro and it worked properly, however, when the incubation was carried out on a larger scale, the degradation factor achieved was 0.45 which is considered lower compared to other studies that evaluated feather hydrolysis In vitro [9,30]. However, scanning microscopy analysis have shown some changes in the pattern of the original feathers.

Both methods, microbial and thermal hydrolysis, showed differences in the concentrations of essential AA, such as lysine, methionine, tryptophan, tyrosine, arginine, and phenylalanine, with some reduction in the sulfur AA in HFMBs. Other feather-degrading studies with keratinolytic strains reported similar results, suggesting that the bacterial biomass improved the content of some essential AA such as lysine, histidine and methionine, and reduction in cystine [27,31].

The quality of protein was estimated according to Friedmam [32]. The PER estimates on feathers meal were high and it is an indicative that protein has important amino acids. All protein sources studied (PER > 2.0) were considered a good quality protein. However, protein quality is a measurement of the AA balance that is absorbed and used for growth and other functions. The total tract digestibility indeed counts for protein quality. The hydrolysis by *Bacillus* spp. tends to improve PER. It may occur due to fermentation process and the microbial biomass formed. In this study, *Bacillus* spp. remained present and viable in the meal after the hydrolysis process. After hydrolysis, feathers were dried at 65 °C, which may have induced sporulation, arousing interest in the probiotic capacity of this strain [10]. Counting viable bacteria in diets and feces revealed that the HFMBs carried *Bacillus* spp. only in dogs fed HFMBs, which ones passed through the dogs’ gastrointestinal tract and were found in viable form within the fecal content, being then possibly active in the host [11]. The resistance of *Bacillus* spp. to the conditions of feed processing and digestive tract must bring out the possibility of this strain to have a probiotic function in dogs, similarly to that suggested for broilers [33]. It is known that it depends on several factors, such as dose administered, period of time, method of application in the diet, frequency of feeding with the probiotic, environmental stress factors, and others [34]. But at least, it could be observed the resistance of the Bacillus to the digestive action in dogs.

The inclusion of feather meals improved the consumption of protein once protein in the diets was increased by addition of HFMBs or HFMT. There were no differences in the consumption of the other dietary components. The lowest total tract digestibility coefficients of organic matter, crude protein, and crude energy were found in the diet with HFMBs. Murray et al. [35] assessing animal by-products, processed or not, as ingredients for dog diets, considered that the presence of feathers is one of the factors that could negatively influence the total tract digestibility of poultry by-products. It is assumed that degradation by microorganisms was not as efficient as the thermal process, as the remaining disulfide bridges in the keratin structure impair the use of protein. It has been recognized that the extensive hydrolysis of keratins requires a combination of proteolytic enzymes and adequate redox environment [36]. Thus, it can be hypothesized that disulfide bonds of the HFMBs were less affected, causing the low solubility of the HFMBs, making it difficult for the digestive enzymes of dogs (pepsin and pancreatic enzymes). Previous studies have evaluated feather meal hydrolyzed by thermal treatment in dog food. Cavalari et al. [37] found an ATTD coefficient of GE, DM and CP as 79.8, 76.0 and 82.3%, respectively. Similar ATTD was found in our study with GE values = 86%, DM = 79% and CP = 85%. Pacheco et al. [5], found that feather meal with enzymes included in the thermal hydrolysis process, under lower pressure and temperature, resulted in an improvement in the digestible energy compared to feather meal without enzymes (77%, 66%, respectively) and concluded that the enzyme-processed ingredient can be considered a source of protein in adult dog diets. Studies with pigs using thermal processed feather meal have shown that it can be used as a protein ingredient as long as supplemented with synthetic AA [38,39,40].

The moisture and fecal score were not affected by the treatments in range considered ideal for the score, according to Moxham [19]. Similar results have been reported by Pacheco et al. [5] which included 7 and 15% of hydrolyzed feather meal in the diets and the dogs produced feces within an appropriate fecal score. Stool production (g/day) was increased by the inclusion of HFMBs (149 g/day), which differed from the control diet (120 g/day) but showed no significant difference in relation to HFMT (134 g/day) (*p* > 0.05). Pacheco et al. [5], also found higher fecal production for the group of dogs fed 15% of feather meal processed in the thermal method or processed with addition of enzymes and under lower temperature in relation to the basal diet (222, 209 and 180 g/day, respectively). As expected, the fecal ammonia content was increased by the inclusion of feather meal, which indicates the presence of the greater amount of undigested protein into the hindgut of dogs fed HFMBs compared to HFMT. This fact represents high availability of protein to the microbiota [41], impacting on fecal characteristics, mainly the concentration of ammoniacal nitrogen in the feces.

The ATTD coefficient value for HFMBs (40%) found in our study shows that under the conditions used, the bioprocess with *B. subtilis* was not able to provide the nutrients and energy as the thermal process. Malmann [42], in a study carried out with feathers in the broilers diet, observed that the lowest coefficient of ileal digestibility was found for the feathers hydrolyzed by *B. subtilis* compared to thermal processed feather meal.

Upon further inspection of the phylum bacteria present in the HFMBs, it was observed that the phylum Firmicutes, genus *Bacillus* was the most abundant, showing that microorganisms remain in the hydrolyzed feather proteins and are able to colonize in the environment (feathers in nature). *Bacillus* are able to synthesize several proteolytic enzymes during their growth or sporulation phase [43], however, in our study, *Bacillus* did not perform this function with requisite efficiency that would significantly improve the digestibility of the ingredient.

Diet is the main factor that modulates the microbial activity in the gastrointestinal tract of dogs. The digesta that reaches the large intestine will determine the microbiota by the amount and type of substrate that it provides [44]. The presence of phyla was similar in the feces of dogs fed diets containing 9.09% HFMBs and 9.09% HFMT. Handl et al. [45] analyzed the fecal microbiota of 12 healthy pet dogs and 12 cats and identified that the most abundant bacteria phylum was Firmicutes, followed by Bacteroidetes in dogs and Actinobacteria in cats. Ritchie et al. [46] and Suchodolski et al. [47] constructed 16S rRNA gene clone libraries to characterize the microbiota in chyme samples from various segments of the gastrointestinal tract of healthy dogs and cats. Firmicutes, Bacteroidetes, and Proteobacteria were the most abundant phyla observed. Similar results were observed in this study, with Bacteroidetes being the most abundant phylum in both treatments. Middelbos et al. [48], obtained similar results when fed dogs with or without fiber supplementation (7.5% of beet pulp) and observed approximately 129 OTU, with Fusobacteria (23–40% of readings), Bacteroidetes (31–34% of readings) and Firmicutes (14–28% of readings).

It is important to highlight that dogs used in this study had the ideal body condition score 5 out 9 points [49] and fecal score, which is comparable with the microbiota present in lean dogs. Previous research has shown a greater abundance of Firmicutes in obese mice fed Western diets, concomitant decreasing abundance of Bacteroidetes [50,51]. Furthermore, some studies have described that the gut microbiota of obese animals and humans exhibits a higher Firmicutes/Bacteroidetes ratio compared with normal-weight individuals [52].

Studies using cultivation methods revealed that *Bacteroides*, *Clostridium*, *Lactobacillus*, *Bifidobacterium*, and Enterobacteriaceae are the predominant bacterial groups in the canine and feline intestine [53,54,55]. However, the most observed genera in the feces of dogs fed 9.09% HFMBs or 9.09% HFMT were *Bacteroides*, *Cetobacterium*, and *Fusobacterium*. Species from the *Bacteroides* genus generally inhabit the colon, are chemoheterotrophic and promotes the catabolism of branched chain amino acids [56]. Amino acid metabolism by gut microbiota results in a complex mixture of metabolic end products including among others ammonia, short-chain and branched-chain fatty acids, which are important bacterial metabolites influencing epithelial physiology and modulating the mucosal immune system of the host [57]. *Bacteroides* spp. help the host by limiting the colonization of pathogens in the gastrointestinal tract [58] through the high production of bacteriocins in the intestine. Bacteriocins are antibacterial substances that reduces bacteria that share the same ecological niche by competition [59].

The presence of the genus *Lactobacillus* in the feces of dog who received HFMBs was observed, which may be related to a probiotic effect of feather meal hydrolyzed by microorganisms, since bacteria of the genus *Bacillus* are potential probiotics and may have provided a better environment for gender development. In the same group, the presence of the genus *Escherichia* was observed, but no cases such as apathy, diarrhea, urinary infection, or any other clinical signs of *E. coli* strains (pathogenic form) were identified during the study.

Both groups of dogs revealed a highly diverse bacterial microbiota in canine feces, as a result of the interaction of the diet with a gut microbiota. Firmicutes was the most abundant phyla in the HFMBs and the most part of them were represented by Bacillaceae with a presence of *Bacillus*, which was expected. By the addition of 9.09% of HFMBs in the diet, *Bacillus* were not so expressive in the fecal samples and a slightly presence of *Lactobacillus* was observed. Also, *Escherichia* was observed in the same treatment, however no negative effects were associated.

## 5. Conclusions

*Bacillus subtilis* FTC01 has high feather-degrading activity in vitro, but its action was limited when applied to a scale-up prototype using increased feather concentration in the cultivation medium. Thus, the degradation was limited, which generated a less digestible HFMBs. In addition, *Bacillus subtilis* proved to be resistant and passed through the gastrointestinal tract remaining viable in the feces and may be able to interfere with the dog’s microbiota. Improving the technique of feather meal hydrolysis by microorganisms can be a way to produce a rich and available ingredient that can still carry bacteria that may have a probiotic effect.

## Figures and Tables

**Figure 1 animals-11-03553-f001:**
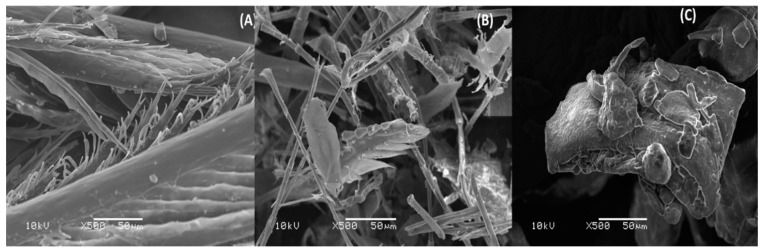
Scanning electron microscopy in natura feather (**A**), *Bacillus subtilis* hydrolyzed feather meal (**B**) and thermal hydrolyzed feather meal (**C**).

**Figure 2 animals-11-03553-f002:**
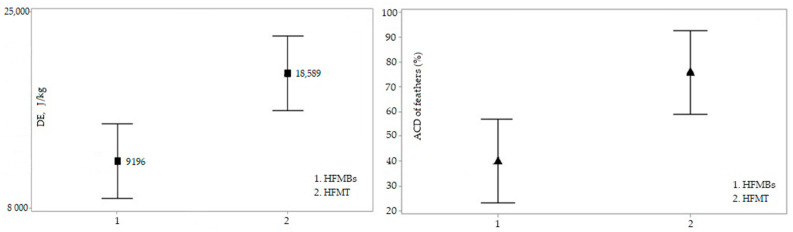
Digestible energy values for HFMBS and HFMT in adult dogs (HFMBS: 9196 J/kg; HFMT: 18,589 J/kg; *p* < 0.05). Different means by the Tukey test 5%. Apparent total tract digestibility values for HFMBs and HFMT energy (HFMBs: 40.0; HFMT: 76.0, *p* < 0.05). Different means by the Tukey test 5%.

**Figure 3 animals-11-03553-f003:**
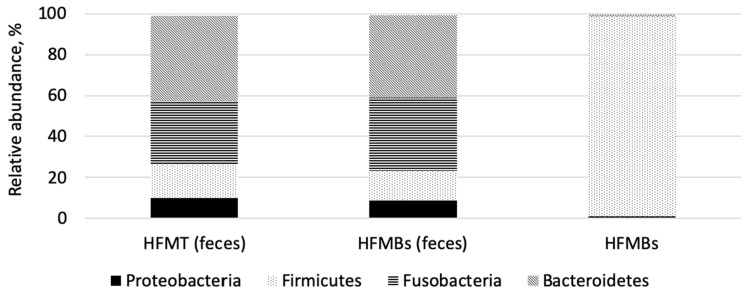
Abundance of Bacteria phyla present in fecal microbiota of dogs consuming diet containing 9.09% *Bacillus subtilis* hydrolyzed feather meal (HFMBs), 9.09% thermally hydrolyzed feather meal (HFMT), and pure *Bacillus subtilis* hydrolyzed feather meal (HFMBs). Phyla showed in the figure represents more than 99% of bacteria identified. The other phyla detected are Tenericutes, Verrucomicrobia, Lentisphaerae, Chloroflexi, Deferribacteres, Epsilonbacteraeota, Acidobacteria, and Actinobacteria.

**Figure 4 animals-11-03553-f004:**
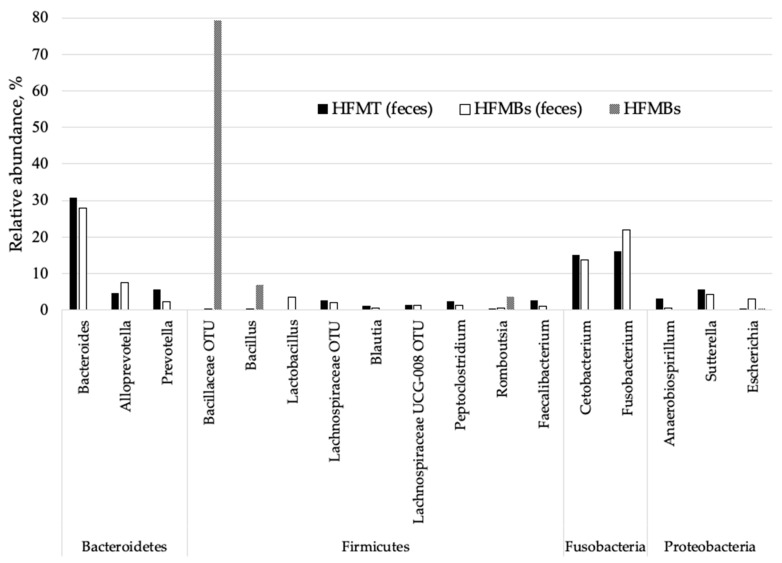
Relative abundance (%) of bacterial genus present in fecal microbiota of dogs consuming diet containing 9.09% *Bacillus subtilis* hydrolyzed feather meal (HFMBs), 9.09% thermally hydrolyzed feather meal (HFMT), and pure *Bacillus subtilis* hydrolyzed feather meal (HFMBs). Genus showed in the figure represents more than 90% of bacteria identified.

**Table 2 animals-11-03553-t002:** Ingredients and chemical composition of experimental diets containing 9.09% feather meal hydrolyzed by *Bacillus subtilis* (HFMBs) and 9.09% thermal feather meal (HFMT).

	Treatments		
	BD ^1^	BD + HFMBs ^2^	BD + HFMT ^3^
Ingredients (g/kg)			
Basal diet	100	100	100
HFMBs	0	10	0
HFMT	0	0	10
Chemical composition (g/kg)			
Dry matter	967	964	964
Mineral matter	88	83	82
Crude protein	271	327	326
Ether extract	130	124	126
Crude fiber	52	47	47
Gross energy, J/kg	20,359	20,602	20,735

^1^ Basal diet (BD)—Ingredients of 1000 kg basal diet: rice grains (244.8 kg); rice brains (229.2 kg); corn (156.3 kg); bovine meat and bone meal (130.2 kg); poultry viscera meal (125.0 kg); wheat bran (104.2 kg); sodium chloride (5.2 g); premix mineral/vitamins (4.2 kg)—adding per kg of diet: Vitamin A. 7000 IU; Vitamin B1. 2 mg; Vitamin B12. 25 mcg; Vitamin B2. 4 mg; Vitamin B6. 2 mg; Vitamin D3. 600 IU; Vitamin E. 50 IU; Vitamin K3. 1 mg; Folic Acid. 0.2 mg; Pantothenic Acid. 10 mg; Biotin. 0.03 mg; Niacin. 30 mg; Cobalt. 10 mg; Copper. 7 mg; Iron. 80 mg; Iodine. 1.5 mg; Manganese. 5 mg; Selenium. 0.2 mg; Zinc. 100 mg; Antioxidant (BHT). 150 mg; yucca (0.3); caramel colorant (0.5). ^2^ DB + HFMBs = basal diet added by 9.09% feather meal hydrolyzed by *Bacillus subtilis*; ^3^ DB + HFMT = basal diet added by 9.09% thermal hydrolyzed feather meal.

**Table 3 animals-11-03553-t003:** Nutrient intake, apparent total tract digestibility, fecal and urinary characteristics of dogs fed diets containing 9.09% feather meal hydrolyzed by *Bacillus subtilis* and 9.09% thermally hydrolyzed feather meal.

Items	Diets ^1^				
	BD	BD + HFMBs	BD + HFMT	SEM ^2^	*p*-Value ^3^
Body weight, kg					
Initial	11.4	11.2	11.2	0.429	0.984
Final	11.5	11.3	11.3		
Daily intake, g/day					
DM	217.5	216.4	225.3	30.22	0.845
OM	198.5	197.8	206.2	27.6	0.828
CP	294.5 ^b^	367.8 ^a^	381.4 ^a^	47.0	0.010
Water consumption, mL/day	442.6	445	439.4	101.8	0.995
Energy intake, J/day					
DE	3820	3839	4176	581,2	0.471
ME	3550	3550	3837	535.1	0.544
Apparent total tract digestibility, %					
DM	79.89	75.46	78.61	2.422	0.013
OM	84.09 ^a^	78.72 ^b^	82.48 ^a^	1.957	0.001
CP	87.81 ^a^	74.81 ^b^	85.02 ^a^	3.052	0.001
Acid hydrolyzed fat	89.37	87.13	87.73	2.745	0.366
Gross Energy	83.96 ^a^	79.61 ^b^	83.09 ^a^	1.907	0.005
Nutritional value of diet, J/ kg					
DE analysed	17,092 ^b^	17,184 ^b^	18,079 ^a^	401.9	0.001
ME estimated ^4^	15,849 ^b^	15.853 ^b^	16.573 ^a^	381.9	0.004
Urinary and fecal characteristics					
Total volume, mL/day	243.4	247.5	297.8	82.41	0.427
pH urinary	7.63	7.41	7.04	0.442	0.083
Urine density, g/L	1.033	1.028	1.033	0.005	0.204
Urine energy, J/day	7.88	7.07	5.47	1.44	0.578
Fecal score ^5^, 1 to 5	2.86	2.82	2.90	0.06	0.725
pH fecal	6.64	6.78	6.84	0.390	0.672
Fecal DM, %	36.63	35.61	34.98	1.522	0.183
Feces, g/day	119.65 ^b^	148.68 ^a^	133.67 ^ab^	17.944	0.041
Feces, g/d (DM g/day)	43.56 ^b^	52.87 ^a^	46.83 ^ab^	6.115	0.05
Amonia ^6^ (DM g/kg)	2.090 ^b^	2.658 ^ab^	3.377 ^a^	0.850	0.045

^1^ BD = diet basal; DB + HFMBs = basal diet added by coverage in 9.09% *Bacillus subtilis* hydrolyzed feather meal; DB + HFMT = basal diet added by coverage in 9.09% thermal hydrolyzed feather meal. ^2^ Standard error mean. *n* = 6 dogs per diet. ^3^ Means followed by superscript different letters (a; b) in the same row differ significantly by Tukey test (*p* < 0.05). ^4^ ME estimated according to the FEDIAF (12). ^5^ Fecal score (19). No significant by the Kruskal-Wallis test. ^6^ Amonia-N (g/kg) = N × correction factor × 17 × (volume of acid − blank)/sample weight (g).

**Table 4 animals-11-03553-t004:** Colony forming units (CFU/g) in the basal diet (BD), diets containing 9.09% (*w*/*w*) feather meal hydrolyzed by *Bacillus subtilis* (HFMBs), 9.09% (*w*/*w*) thermal hydrolyzed feather meal (HFMT) and feces that consumed the diets.

Diet	Bacterial Counts (CFU/g)
BD	<10
HFMT	<10
HFMBs	5.9 × 10^4^
**Feces**	
Dogs that received HFMBs	
A ^1^	1.19 × 10^5^
B ^1^	1.09 × 10^6^
C ^1^	1.76 × 10^6^
D ^1^	5.30 × 10^6^
E ^1^	1.71 × 10^7^
Dogs that received HFMT ^2^	<10
Dogs that received DB ^2^	<10

^1^ The amount obtained in each subject. ^2^ Mean value in each diet, *n* = 6. Value < 10 means below the detection limit.

## Data Availability

No new data were created or analyzed in this study. Data sharing is not applicable to this article.

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
