# Peer review of "Determination of the Nutritional Value of Diet Containing Bacillus subtilis Hydrolyzed Feather Meal in Adult Dogs"

_animals, 2021, doi:10.3390/ani11123553_

Round 1
Reviewer 1 Report
The study concerns an important area for improving utilization of feather as an ingredient in pet food. Critics have been rised towards the pet food industry for applying human grade ingredients. Utilization of feathers as protein ingredient in dog food is therefore a contribution to a more sustainable pet food production.
The study is very well conducted and the discussion is relevant. The English language mainly excellent. Here are some minor comments on language and errors:
Line 21: ...on it. Please describe expected effects.
Table 1: Third column feather meal for DM and CP is not correct. Total aas should be explained.
General: the terms "digestibility" and "ileal digestibility" is used. I would prefer "faecal digestibility" or "total tract digestibility" in stead of "digestibility" make the difference clear.
Table 3. Amonia in the last row.
Author Response
Dear Reviwer,
thanks for revising the manuscript,
We made several modifications on the manuscript. Your suggestions were attended.
The points between number were missed in the third column of Table 1. However the column was deleted.
The term "digestibility" was exchanged by "Apparent Total Tract Digestibility""
The term "amonia" was fixed.
Best regards,
LUciano
Reviewer 2 Report
It is overall interesting paper.Please find the following line edits:Page1 L40:Use singular words in the key words and replace “,” with “;”Page4 L53: Delete the abbreviation and replace “2.4.1.” with “2.4.1”
Page9 L4: The title should be brief and related information should exist in the table notes
Page9 Table3: Delete ns
Page12 L10: Delete and
Page12 L17: Replace “, decreasing” with “and decrease”
Page13 L18: Delete “and better characteristics of meat”
Discussion needs to be revised to make it more logical
Author Response
Dear Reviewer,
Thanks for all he comments on the manuscript.
Several modifications on the manuscript has been made according to all the reviewers.
We tried to replace pural word by singular ones and we attended to your comments.
Best regards,
Luciano Trevizan
Reviewer 3 Report
The manuscript has severe shortcomings. Material and methods should be more precisely described, especially which formulas were used to calculate the nutrional values. The discussion does not come to the point and does not reflect the outcome of the study. The outcome is a feedstuff with a digestibility of 40 % but the conclusion implicates that using Bacillus would be beneficial to use. It is unclear if the investigated feed samples are coming from the same source (batch)
Furthermore, in several cases the results are repeated in the discussion. The discussion could be written more straight forwarded. In relation to the outcome of the study the discussion appears too long .
In addition English language use and style does not meet the standards in my opinion.
Throughout the entire manuscript please use SI Units when expressing energy values (Joule instead of calories)!!
Line 56 keratin biodegradation by microorganism….please be more specific…how many nutrients are lost during fermentation with Bacillus….What are the endproducts of fermentation and nutrient losses at the end of fermentation cycle…Feathers mainly consist of amino acids and these may be degraded as well, did you ever measure biogenic amines or ammonia in the final fermentation productto see how much was actually degraded…Fermentation alays have nutrient losses please explain how much losses occur !!
Line 90 …light dark cycle of 14h:10h…not 10 min or???
Line 126 to 140…It is completely unclear if conventional feather meal is from the same batch as Bacillus treated fetaher meal.
Line 94 What is the use of the feathers in natura…These data are not obtained from the original feather meal batch used in this experiment…it is from the literature from 1986…feathers in natura can be deleted
Table 1 It seems to me that the differences in Amino acid compostion of HFMB and HFMC are within the range of usual variation of feathermeal in Brazil …please inspect amino acid composition table in Amino Dat5.0 by Evonik…As it seems that the feathers are not coming from the same batch this could be the usual variation in amino acid composition of the feather meal and not an effect by bacillus fermentation. Did you measure fermentation endproducts in HFMBs, without these values i it is unclear if there was any fermentation at all. What is the source of energy for the Bacillus, if they have to grow and change the amino acid composition of feather meal??
General remark…What about hygienic status of the feather meals…high temperature hydrolyzation reduces pathogenic bacteria while low temperature fermentation is less effective for hygienic sanitation.
Line 170-173 Why calculating three different PER, the results should be more or less the same, please use the most relevant equation and delete the other ones. There is no explanation in the discussion why the results for different PER equation differ
Line 201…should be (day 6 to day 10)…as it were two blocks of 10 d (line 199)
Line 212 I hope you used a graduated cylinder. Beakers are not correctly graduated.
Line 238…Becker..???
Line 288 see previous comments…are samples from the same batch??
Footnote 1 and 2 below Table 1 can be deleted…is described in the material and method section
Table 2. Sorry, as you did not reduce the basal diet to 90 g, you added 9.09 % of the different feather meals to the diet!! Please rewrite. Please change throughout the manuscript.
Table 3 You called the manuscript ..Determination of Nutritional value of …hydrolyzed feather meal…but you only measured the ME and DE of the diet and not of the feather meal. What was the purpose of this research the diets or the feather meal. For detreminationation oft he feather meal you have calculate the DE and ME fort he feathermeal using the difference method. Please explain more precisely the calculation of digestibilities in the material and method section…Adeola and FEDIAF gives several formulas in the book chapter cited.
Line 326 Figure 3 how were these values calculated?? A digestibility of 40 % does not look very promising. Is there any value given oft he digestibility of feathers when not treated , (as you call it in natura)!!!
Table 4 why are the individual bycterial counts for HFMBs given for but fort he other diets the average oif all animals …please change give the average for each group, you can always include the standard deviation
Please rewrite discussion… you don‘t have to repeat the results of your own study each time…for example line 405, 468-470, etc
Line 390how much bacterial biomass is produced during fermentation and if the digestibility is as low as 40 % the animal has no use of the amino acids …the results and the discussion should come to the point and should not overemphasizing results of minor importance
Line 422 higher protein consumption…not again ist he result of notformulating isonitrogenous diets
435 please specify …conventional hydrolyzed feather meal“
Line 445 The digestibility implies in the feces quality??? What does that mean. I do not understand.
Line 456 greater amount of indigestible protein reaching the hindgut …rewrite the gut is the entire gastrointestinal tract
Line 460- 463 does not fit to your study..No diets with feathermeal in Reference 42
Line 493…why ist he ideal body condition score for this reaerch of interest, by the way these values wer not given.
Line 494-496 Mouse research….in obese rats…do you think that the microbiom of dogs and rats is comparable??? And what part of the intestinal tract do you mean
Line 502-508 How is this related to your own research ..you added feather proteins and not complex sugars and carbohydrates…
The conclusion should indicate that fementation by Bacillus is not useful as digestibility is too low compared toconventional hydrolyzed feathers. Furthermore, what about cost and time of this process. If I want to feed a probiotic, I can choose a probiotic and not a low digestible feed. Which contains a probiotic. Please come to a conclusion which reflect your results
Author Response
Dear Reviewer,
Thanks so much for reviewing this manuscript.
Follow the answers to the revisor 3 and the text with track changes.
All the comments are in the manuscript and follow questions and answers.
Best regards
Luciano Trevizan
The manuscript has severe shortcomings. Material and methods should be more precisely described, especially which formulas were used to calculate the nutritional values. The discussion does not come to the point and does not reflect the outcome of the study. The outcome is a feedstuff with a digestibility of 40 % but the conclusion implicates that using Bacillus would be beneficial to use. It is unclear if the investigated feed samples are coming from the same source (batch)
Furthermore, in several cases the results are repeated in the discussion. The discussion could be written more straight forwarded. In relation to the outcome of the study the discussion appears too long.
In addition English language use and style does not meet the standards in my opinion.
Throughout the entire manuscript please use SI Units when expressing energy values (Joule instead of calories)!!
All the information about calories were expressed in Joules.
Line 56 keratin biodegradation by microorganism….please be more specific…how many nutrients are lost during fermentation with Bacillus….What are the endproducts of fermentation and nutrient losses at the end of fermentation cycle…Feathers mainly consist of amino acids and these may be degraded as well, did you ever measure biogenic amines or ammonia in the final fermentation product see how much was actually degraded… Fermentation always have nutrient losses please explain how much losses occur !!
During feather fermentation, Bacillus (and other microorganisms) use the amino acids resulting from keratin hydrolysis as C, N and energy source. Therefore, part of the amino acids present in the original feather keratin will be degraded to support microbial growth.
A sentence was added: “even through there is loss of nutrients during the fermentation”
Line 90 …light dark cycle of 14h:10h…not 10 min or???
Revised as indicated
Line 126 to 140…It is completely unclear if conventional feather meal is from the same batch as Bacillus treated feather meal.
Lines 138-140. It is clearly described that five bioreactor baths were prepared and then homogenized to form a single meal used in the experiment.
Line 125, “…and 50 kg of the meal originated from the same batch was separated…”
Line 94 What is the use of the feathers in natura…These data are not obtained from the original feather meal batch used in this experiment…it is from the literature from 1986…feathers in natura can be deleted
The values for DM and CP were determined in this work. The value of DM was expressed in g/kg and it was changed to % according to the legend
Data of feather in natura was deleted.
Figure 2, about PER equations was deleted and results from only one equation was described in the text.
Table 1 It seems to me that the differences in Amino acid compostion of HFMB and HFMC are within the range of usual variation of feathermeal in Brazil …please inspect amino acid composition table in Amino Dat5.0 by Evonik…As it seems that the feathers are not coming from the same batch this could be the usual variation in amino acid composition of the feather meal and not an effect by bacillus fermentation. Did you measure fermentation endproducts in HFMBs, without these values i it is unclear if there was any fermentation at all. What is the source of energy for the Bacillus, if they have to grow and change the amino acid composition of feather meal??
Feather meal used in the experiment was from a same batch, thus differences could not be attributed to variations from feathers. Part of the amino acids used to support microbial growth should be derived for synthesis of proteins and other essential N-containing compounds. Cultivation of Bacillus was not under anaerobic conditions, thus amino acids should be metabolized through oxidative deamination followed by catabolism of the resulting C backbones (the source of energy for Bacillus).
Biosynthesis of a bacterial cell, with organic compounds as sources of energy and carbon, requires approximately 20 to 60 billion high-energy phosphate bonds. A substantial fraction of this energy budget is devoted to biosynthesis of amino acids, the building blocks of proteins. The fueling reactions of central metabolism provide precursor metabolites for synthesis of the 20 amino acids incorporated into proteins. Thus, synthesis of an amino acid entails a dual cost: energy is lost by diverting chemical intermediates from fueling reactions and additional energy is required to convert precursor metabolites to amino acids (Akashi & Gojobori, PNAS 2002, 99 (6) 3695-3700; https://doi.org/10.1073/pnas.062526999). It seems therefore that providing a growth medium rich in amino acids would be energetically favorable for microbial growth (at least in terms of protein synthesis, which is largely energy-demanding).
General remark…What about hygienic status of the feather meals…high temperature hydrolyzation reduces pathogenic bacteria while low temperature fermentation is less effective for hygienic sanitation.
Agree, microbiological evaluation of the fermentation product would be interesting (I thought it had already been done, and the total counts, coliforms, molds and yeast values were in agreement with the standards). However, Bacillus are recognized to produce antimicrobial compounds and due to their metabolic abilities and be inoculated at high counts are often successful to exclude potential pathogens by competition. The idea is similar to the use of a starter culture for conventional food fermentation.
Line 170-173 Why calculating three different PER, the results should be more or less the same, please use the most relevant equation and delete the other ones. There is no explanation in the discussion why the results for different PER equation differ
We selected the equations that explain better the results. Additional information was added.
Line 201…should be (day 6 to day 10)…as it were two blocks of 10 d (line 199)
Ok, changed as indicated
Line 212 I hope you used a graduated cylinder. Beakers are not correctly graduated.
Line 230 - It has been a translation issue; samples were measured in a graduated cylinder.
(Google translates “proveta” as “beaker”)
Line 238…Becker..???
Revised to “flask”
Line 288 see previous comments…are samples from the same batch??
See previous answer
Footnote 1 and 2 below Table 1 can be deleted…is described in the material and method section
Ok, agree. It was deleted.
Table 2. Sorry, as you did not reduce the basal diet to 90 g, you added 9.09 % of the different feather meals to the diet!! Please rewrite. Please change throughout the manuscript.
Thanks. That is completely through. It was changed accordingly throughout the text.
Table 3 You called the manuscript... Determination of Nutritional value of …hydrolyzed feather meal…but you only measured the ME and DE of the diet and not of the feather meal. What was the purpose of this research the diets or the feather meal. For determination of the feather meal you have calculate the DE and ME for the feather meal using the difference method. Please explain more precisely the calculation of digestibilities in the material and method section…Adeola and FEDIAF gives several formulas in the book chapter cited.
The title was changed to: Determination of the Nutritional Value of Diet Containing Bacillus subtilis Hydrolyzed Feather Meal in Adult Dogs
Formula has been added.
Line 326 Figure 3 how were these values calculated?? A digestibility of 40 % does not look very promising. Is there any value given of the digestibility of feathers when not treated, (as you call it in natura)!!!
Agree, 40% does not seems very promising. But last study that we tested digestibility of thermal processed feather meal it was close to 67% (https://doi.org/10.1590/S1806-92902016000600002). No studies in dogs feeding raw feathers was found to compare results. In vitro digestibility of feathers must be close to 9.6% (https://doi-org.ez45.periodicos.capes.gov.br/10.1016/j.anifeedsci.2005.06.002).
Table 4 why are the individual bacterial counts for HFMBs given for but for the other diets the average of all animals …please change give the average for each group, you can always include the standard deviation
The total counts were always lower that 10 colonies (below the detection limit).
Dogs eating HFMBs had consistent difference in the fecal Bacillus, then results were expressed for each dog separately.
Please rewrite discussion… you don‘t have to repeat the results of your own study each time…for example line 405, 468-470, etc
Adjust was made.
Line 390 how much bacterial biomass is produced during fermentation and if the digestibility is as low as 40 % the animal has no use of the amino acids …the results and the discussion should come to the point and should not overemphasizing results of minor importance
Unfortunately, it is very hard to determine bacterial biomass using solid substrates like feathers. In particular, the mentioned study suggested the improvement of some essential amino acids is due to the presence of bacterial biomass in the final product.
Line 422 higher protein consumption…not again is the result of not formulating isonitrogenous diets
Ok. Deleted.
435 please specify …conventional hydrolyzed feather meal“
We assumed instead of conventional, Thermally hydrolyzed feather meal (HFMT);
Line 445 The digestibility implies in the feces quality??? What does that mean. I do not understand.
It was a mistake. It was deleted.
Line 456 greater amount of indigestible protein reaching the hindgut …rewrite the gut is the entire gastrointestinal tract
It is already described as “large intestine” in the text.
Line 460- 463 does not fit to your study.. No diets with feather meal in Reference 42
Ok, this part was deleted.
Line 493…why is the ideal body condition score for this reaerch of interest, by the way these values were not given.
It has been added in the same sentence (5 out 9 points).
Line 494-496 Mouse research….in obese rats…do you think that the microbiom of dogs and rats is comparable??? And what part of the intestinal tract do you mean
In fact, the Firmicutes to Bacteroidetes ratio has been associated with obesity in mammals. Despite many studies have been conducted in mice and rats, there are studies for other animals, also human studies. Maybe it should be included a sentence indicating that similar results have been described for other mammals.
“Some studies have described that the gut microbiota of obese animals and humans exhibits a higher Firmicutes/Bacteroidetes ratio compared with normal-weight individuals, proposing this ratio as an eventual biomarker.”
Magne F, Gotteland M, Gauthier L, Zazueta A, Pesoa S, Navarrete P, Balamurugan R. The Firmicutes/Bacteroidetes ratio: A relevant marker of gut dysbiosis in obese patients? Nutrients. 2020;12(5):1474. doi:10.3390/nu12051474
Cui C, Shen CJ, Jia G, Wang KN. Effect of dietary Bacillus subtilis on proportion of Bacteroidetes and Firmicutes in swine intestine and lipid metabolism. Genet Mol Res. 2013, 12(2):1766-76. doi: 10.4238/2013.
Line 502-508 How is this related to your own research ..you added feather proteins and not complex sugars and carbohydrates…
Agree, it could be revised to the amino acid metabolism. The fact that Bacteroides are chemoheterotrophic is relevant. We have added a sentence.
Neis EP, Dejong CH, Rensen SS. The role of microbial amino acid metabolism in host metabolism. Nutrients. 2015;7(4):2930-2946. doi:10.3390/nu7042930
Yoshida N, et al. Bacteroides spp. promotes branched-chain amino acid catabolism in brown fat and inhibits obesity. iScience, 2021, 24(11):103342. doi: 10.1016/j.isci.2021.103342.
The conclusion should indicate that fementation by Bacillus is not useful as digestibility is too low compared to conventional hydrolyzed feathers. Furthermore, what about cost and time of this process. If I want to feed a probiotic, I can choose a probiotic and not a low digestible feed. Which contains a probiotic. Please come to a conclusion which reflect your results
Ok, in my opinion it is stated in the discussion that the Bacillus had not significant effect on improving digestibility. The discussion was reduced with focus on most relevant points;
Thank you so much for your expertise and for pointing out observations and comments on the manuscript.
Best regards
Luciano Trevizan
Round 2
Reviewer 2 Report
no comments
Reviewer 3 Report
no further comments